# Reasons for Discontinuing Treatment with Sodium-Glucose Cotransporter 2 Inhibitors in Patients with Diabetes in Real-World Settings: The KAMOGAWA-A Study

**DOI:** 10.3390/jcm12226993

**Published:** 2023-11-09

**Authors:** Yuto Saijo, Hiroshi Okada, Shinnosuke Hata, Hanako Nakajima, Nobuko Kitagawa, Takuro Okamura, Takafumi Osaka, Noriyuki Kitagawa, Saori Majima, Takafumi Senmaru, Emi Ushigome, Naoko Nakanishi, Masahide Hamaguchi, Michiaki Fukui

**Affiliations:** 1Department of Endocrinology and Metabolism, Graduate School of Medical Science, Kyoto Prefectural University of Medicine, Kyoto 602-8566, Japan; y-saijo@koto.kpu-m.ac.jp (Y.S.); hatashin@koto.kpu-m.ac.jp (S.H.); tabahana@koto.kpu-m.ac.jp (H.N.); nobuko-s@koto.kpu-m.ac.jp (N.K.); d04sm012@koto.kpu-m.ac.jp (T.O.); tak-1314@koto.kpu-m.ac.jp (T.O.); nori-kgw@koto.kpu-m.ac.jp (N.K.); saori-m@koto.kpu-m.ac.jp (S.M.); semmarut@koto.kpu-m.ac.jp (T.S.); emis@koto.kpu-m.ac.jp (E.U.); naoko-n@koto.kpu-m.ac.jp (N.N.); mhama@koto.kpu-m.ac.jp (M.H.); michiaki@koto.kpu-m.ac.jp (M.F.); 2Department of Diabetes & Endocrinology, Japanese Red Cross Kyoto Daiichi Hospital, Kyoto 605-0981, Japan; 3Department of Endocrinology and Diabetology, Ayabe City Hospital, Kyoto 623-0011, Japan; 4Department of Diabetology, Kameoka Municipal Hospital, Kyoto 621-0826, Japan

**Keywords:** sodium-glucose cotransporter 2 (SGLT2) inhibitors, diabetes, drug discontinuation

## Abstract

Sodium-glucose cotransporter 2 inhibitors (SGLT2is) are a class of antidiabetic agents known to exert cardioprotective, renoprotective, and hypoglycemic effects. However, these agents have been associated with adverse effects, such as genital infection, volume depletion, hypoglycemia, and diabetic ketoacidosis, resulting in drug discontinuation. Herein, we aimed to determine the reasons for discontinuing treatment with SGLT2is among Japanese patients with diabetes. This retrospective cohort study enrolled 766 patients with diabetes who had initiated SGLT2is between January 2014 and September 2021. The follow-up period was 2 years from the initiation of the SGLT2is. Overall, 97 patients (12.7%) discontinued the SGLT2is during the follow-up period. The most common reasons for discontinuing the SGLT2is were frequent urination (19.6%), followed by genital infection (11.3%), improved glycemic control (10.6%), and renal dysfunction (8.2%). A comparison of the characteristics between the continuation and the discontinuation group was conducted, excluding those who discontinued the SGLT2is because of improved glycemic control. The patients in the discontinuation group (68 [55–75] years) were older than those in the continuation group (64 [53–71] years; *p* = 0.003). Importantly, we found no significant association between diabetes duration, diabetic control, renal function, or complications of diabetes in both groups. This real-world study revealed that frequent urination was the most common reason underlying SGLT2i discontinuation among Japanese patients with diabetes. To avoid discontinuation, precautions against various factors that may cause frequent urination must be implemented.

## 1. Introduction

Over the last several decades, the prevalence of diabetes has steadily increased owing, in part, to over-nutrition, a lack of physical activity, and aging societies. As diabetes and its complications and comorbidities gain widespread momentum, healthcare costs are progressively rising worldwide. Therefore, it is crucial to focus not only on glycemic control but also on complications and comorbidities when treating patients with diabetes. SGLT2is were originally developed for the treatment of Type 2 diabetes, some of which were later approved for use in Type 1 diabetes. SGLT2is reduce blood glucose concentration, independent of insulin secretion or sensitivity, by hindering renal glucose reabsorption in the proximal tubule, subsequently increasing urinary glucose excretion.

Sodium-glucose cotransporter 2 inhibitors (SGLT2is) are a relatively new class of antidiabetic agents with cardioprotective, renoprotective, and hypoglycemic effects. SGLT2is reportedly exert beneficial effects against glycemia, high blood pressure, high body weight [1], intrarenal hemodynamics, and albuminuria [2]. In addition, SGLT2is potentially reduce the risk of cardiovascular disease [3,4,5,6] and prevent the progression of chronic kidney disease [7,8,9]. Although the cardioprotective mechanisms of SGLT2is have not yet been fully explained, promising hypotheses include an improvement in ventricular loading conditions induced by the diuretic effects, the enhancement of cardiac metabolism, the inhibition of Na^+^/H^+^ exchange, and so on [10]. The possible molecular biological explanations for their renoprotective mechanisms include the normalization of tubuloglomerular feedback, induced by inhibiting the reabsorption of sodium chloride from the proximal tubule and the subsequent increase in sodium chloride delivered to the macula densa, enhanced mitochondrial function, and reduced oxidative stress [11]. These versatile properties have made SGLT2is rank higher in the treatment of diabetes, heart failure, and chronic kidney disease. Despite their almost “impeccable” pharmacological profile, SGLT2is have been associated with adverse effects, ranging from non-serious to serious, including genital infection, volume depletion, hypoglycemia, and diabetic ketoacidosis, some of which necessitate the discontinuation of SGLT2i therapy [3,12,13]. To obtain maximal benefits, uninterrupted SGLT2i therapy is essential. However, few studies based on real-world data have explored the discontinuation of SGLT2is among patients with diabetes. In addition, the potential measures that could overcome drug discontinuation need to be explored. Accordingly, we used real-world data to determine the reasons for discontinuing treatment with SGLT2is.

## 2. Materials and Methods

### 2.1. Study Design and Patients

This retrospective cohort study was performed using data from the KAMOGAWA-A cohort. The KAMOGAWA-A cohort study, initiated in 2014, is an ongoing prospective study focusing on patients with diabetes. Specifically, this research encompasses outpatients from the Kyoto Prefectural University of Medicine and Kameoka Municipal Hospital who were prescribed SGLT2is between January 2014 and September 2021.

### 2.2. Data Collection

In the KAMOGAWA-A cohort study, blood assessments and urinalysis were performed at every visit. Data concerning blood evaluations, urinalysis, and microvascular complications of diabetes, specifically diabetic neuropathy, retinopathy, and nephropathy, as well as cardiovascular diseases, including stroke, coronary artery disease, and peripheral artery disease, were obtained from the KAMOGAWA-A cohort database. Diabetes was diagnosed according to the Report of the Expert Committee on the Diagnosis and Classification of Diabetes Mellitus [14]. Participants were categorized as insulin users or non-users. Participants were categorized as never-smokers, past-smokers, or current-smokers. Body mass index (BMI) was calculated by dividing weight in kilograms by height in meters squared. The blood examination data on SGLT2i initiation were defined as baseline data. The estimated glomerular filtration rate (eGFR) was calculated using the GFR estimation formula for Japanese individuals: eGFR (mL/min per 1.73 m^2^) = 194 × Cre^−1.094^ × age^−0.287^ (×0.739 for women). The follow-up period was 2 years from the initiation of SGLT2is (dapagliflozin, canagliflozin, luseogliflozin, empagliflozin, ipragliflozin, and tofogliflozin), and the primary outcome was discontinuation during the follow-up period. Discontinuation of SGLT2i was defined as discontinuation within 2 years of SGLT2i initiation. The reasons for the discontinuation of the SGLT2i were collected from the medical records.

### 2.3. The Definition of Microvascular and Macrovascular Complications

Neuropathy was defined by the diagnostic criteria for diabetic neuropathy proposed by the Diagnostic Neuropathy Study Group in Japan. To put it concisely, in the absence of peripheral neuropathies other than diabetic neuropathy in diabetic patients, diabetic neuropathy is diagnosed by two or more abnormalities of three neurological examination items, such as sensory symptoms, diminished or absent ankle reflex (bilateral), and decreased vibratory sensation on the bilateral medial malleoli evaluated using a C128 Hz tuning fork. Retinopathy was assessed using an ophthalmoscope and was categorized as no diabetic retinopathy, simple diabetic retinopathy, or proliferative diabetic retinopathy. Nephropathy was graded as follows: stage 1 (normoalbuminuria), urinary albumin excretion less than 30 mg per gram of creatinine (mg/g Cr); stage 2 (microalbuminuria), 30 to 300 mg/g Cr; stage 3 (macroalbuminuria), more than 300 mg/g Cr; stage 4, eGFR less than 30 mL/min per 1.73 m^2^; or stage 5, the patients with dialysis. Urinary albumin excretion was evaluated using an immunoturbidimetric assay. Cardiovascular disease was defined as the presence of previous myocardial infarction, cerebral infarction, or peripheral artery disease based on the clinical history or physical examination.

### 2.4. Inclusion and Exclusion Criteria

A total of 1028 patients who were prescribed SGLT2is between January 2014 and September 2021 were included in this study. The cases in which the date of SGLT2i initiation was unknown or untraceable due to referral from another hospital were excluded (n = 210). The patients in the continuation group whose follow-up period was less than 2 years were also excluded (n = 52).

### 2.5. Ethics

This study was approved by the Research Ethics Committee of the Kyoto Prefectural University of Medicine (No. ERB-C-1876) and was conducted in accordance with the principles of the Declaration of Helsinki. Informed consent was obtained in the form of opt-out. Patients who were unwilling to participate in the study were excluded.

### 2.6. Statistical Analysis

Continuous variables were expressed as median (interquartile range). The differences in characteristics between the continuation and discontinuation groups were evaluated using the Mann–Whitney U or chi-square tests. A comparison of the characteristics between the continuation and the discontinuation groups was conducted, excluding those who discontinued the SGLT2is because of improved glycemic control. Differences were considered statistically significant at *p* < 0.05. Data analyses were performed using the JMP ver. 17 (SAS Institute Inc., Cary, NC, USA).

## 3. Results

Table 1 presents the characteristics of all patients enrolled in the current study. The data regarding characteristics were collected when the SGLT2is were initiated. We enrolled 766 patients, of whom 441 (57.6%) were males and 53 (6.9%) had been diagnosed with type 1 diabetes; moreover, 175 (22.8%) were insulin users. The median age was 64 [54–71] years old. The median duration of diabetes was 11 [4–18] years. The median body mass index was 25.3 [23.0–28.3] kg/m^2^. The median hemoglobin A1c (HbA1c) was 7.9 [7.1–8.7]%. The frequency of each SGLT2i administration is shown in Table 1.

Table 2 presents the reasons underlying the discontinuation of the SGLT2i and their stratification by duration of the SGLT2i treatment. The overall discontinuation rate was 12.7% (97 patients). The number of cases in which the SGLT2i treatment was discontinued within 3 months, 3 to 12 months, and 12 to 24 months was 22 (22.7%), 43 (44.3%), and 32 (33.0%), respectively. The most common reason for SGLT2i discontinuation was frequent urination (19.6%, 19 patients), followed by genital infection (11.3%, 11 patients), improved glycemic control (10.6%, 10 patients), renal dysfunction (8.2%, 8 patients), and urinary tract infection (7.2%, 7 patients). Two out of three patients who discontinued the SGLT2is because of diabetic ketoacidosis had type 1 diabetes. No cases of euglycemic diabetic ketoacidosis were reported. Regardless of the duration of the SGLT2i treatment, frequent urination was the leading cause of discontinuation. The number of patients who discontinued the SGLT2is because of fatigue and urinary tract infection tended to increase within 3 months of and later than 3 months after the SGLT2i initiation, respectively.

Table 3 presents the reasons underlying the discontinuation of the SGLT2i and their stratification by duration of the SGLT2i treatment in patients with type 2 diabetes. In this study, 713 patients with type 2 diabetes were included, among whom 90 patients (12.6%) discontinued SGLT2i treatment. Importantly, this discontinuation rate closely mirrored that of the entire group. Furthermore, no significant variations were found in the reasons for discontinuation when compared to the whole group, the predominant reasons being frequent urination, genital infection, renal dysfunction, and urinary tract infection.

Table 4 delineates the comparison between the groups who continued and discontinued the SGLT2i treatment. Specifically, patients who discontinued the SGLT2is due to improved glycemic control were omitted from this table, as this outcome could potentially represent a favorable effect of the SGLT2i treatment.

Importantly, patients in the discontinuation group were older than those in the continuation group (64 [53–71] vs. 68 [55–75] years old, *p* = 0.003). We found no significant associations involving the duration of diabetes (11 [4–18] vs. 9 [4–18] years, *p* = 0.54), type of diabetes (Type 1/Type 2 diabetes, 46/623 vs. 7/80 patients, *p* = 0.832), glycemic control (HbA1c 7.9 [7.1–8.7]% vs. 7.8 [7.3–8.9]%, *p* = 0.467, blood glucose (155 [127–198] vs. 172 [126–207.5] mg/dL, *p* = 0.096), renal function (eGFR 74.8 [61.0–86.8] vs. 69.9 [57.7–83.9] mL/min/1.73 m^2^, *p* = 0.194), use of insulin (+/−, 516/153 vs. 66/21 patients, *p* = 0.787), or major complications of diabetes.

## 4. Discussion

Our study aimed to investigate the reasons for discontinuing SGLT2i treatment in the examined cohort, thereby elucidating strategies to minimize the discontinuation rate and thus maximize the long-term beneficial effects of SGLT2is. Based on the findings of our real-world study, approximately one-eighth of patients discontinued treatment with SGLT2is, and the most common reason for discontinuation was frequent urination, followed by genital infection, renal dysfunction, and urinary tract infection.

Large-scale clinical trials have assessed the adverse effects associated with SGLT2is, consistently reporting that the genital infection rate was substantially higher in the SGLT2i-treated group than in the placebo group [3,12,13]. In addition, an increase in the number of amputation and acute kidney injury cases has been documented with SGLT2i treatment [12]. Although a segment of the CANVAS Program identified osmotic diuresis as an adverse event of interest, the other trials did not categorize frequent urination as an adverse event, potentially for two specific reasons. Firstly, frequent urination is considered a subjective symptom, and it might be categorized into or expressed as other events that can be observed objectively (e.g., volume depletion, acute kidney injury, and urinary tract infection). Secondly, frequent urination itself does not immediately have serious consequences and is partly attributed to the initial diuretic effect mediated by the SGLT2is [15].

If we consult larger studies, post-marketing surveillance research on SGLT2 inhibitors has consistently reported that frequent urination, genital infection, and urinary tract infection are major adverse drug reactions (ADRs). The post-marketing surveillance research on canagliflozin showed that among 12,227 patients, 1836 ADRs were reported, of which 170 cases (1.39%) were associated with volume depletion, 164 (1.34%) were genital infections, 150 (1.23%) were associated with polyuria and pollakiuria, and 145 (1.19%) were urinary tract infections [16]. The research on empagliflozin indicated that among 7931 patients, 1024 ADRs were reported, of which 102 cases (1.29%) were excessive urination or frequent urination, 85 (1.07%) were urinary tract infections, 52 (0.66%) were genital infections, and 40 (0.50%) were volume depletion [17]. The research on ipragliflozin showed that among 11,051 patients, 2107 ADRs were reported, of which 612 cases (5.54%) were polyuria or pollakiuria, 241 (2.18%) were volume-depletion-related events including dehydration, 169 (1.53%) were urinary tract infections, and 158 (1.43%) were genital infections. Interestingly, it also compared the treatment-naïve patients (patients who had not been administered other antidiabetic drugs before initiating ipragliflozin monotherapy) with non-naïve patients. It revealed that treatment-naïve patients had a significantly lower incidence of any ADR, polyuria/pollakiuria, and volume-depletion-related events, which can be partly attributed to the difference in the severity of diabetes between the two groups [18]. The research on tofogliflozin indicated that, among 6711 patients, 846 ADRs were reported, of which 135 cases (2.01%) were volume-depletion-related disorders, 117 (1.74%) were genital infections, 91 (1.36%) were urinary tract infections, and 90 (1.34%) were polyuria/pollakiuria. In addition, it showed that volume-depletion-related disorders and polyuria/pollakiuria were more frequent in patients aged 65 or over than in younger patients [19]. Importantly, this does not necessarily imply that the ADRs reported in these studies led to the discontinuation of SGLT2i treatment. Notably, our study is novel in that it offers new insights into which adverse reactions are more likely to result in the discontinuation of SGLT2i therapy.

A meta-analysis showed that urinary tract infection was associated with dapagliflozin but not with other SGLT2is [20]. In this study, seven patients discontinued SGLT2i treatment due to urinary tract infections; notably, five were administered dapagliflozin, while two received luseogliflozin. However, as this study was retrospective, this result does not prove a causal relationship.

SGLT2is are linked to improvements in diabetic kidney disease (DKD) [21]. DKD is defined as the presence of albuminuria (urinary albumin excretion of more than 30 mg/g Cr), and/or reduced eGFR (<60 mL/min/1.73 m^2^). DKD is the leading cause of end-stage renal disease (ESKD) [22,23], and early diagnosis and intervention can delay the progression of DKD [24]. This underscores the importance of continuing SGLT2i therapy, considering that SGLT2is have renoprotective effects. As for renoprotective effects, the DAPA-CKD trial, the composite primary outcome of which was the first occurrence of any of the following: a decrease of 50% in eGFR, the onset of ESKD, or death from renal or cardiovascular diseases, showed that, compared with a placebo, dapagliflozin significantly reduced the occurrence of a primary outcome (hazard ratio, 0.61; 95% confidence interval [CI], 0.51 to 0.72; *p* < 0.001). Combined with the results of the DECLARE-TIMI58, CANVAS, and EMPA-REG OUTCOME trials, these trials firmly imply that SGLT2is have renoprotective effects.

Notably, instances of decreased renal function in this study encompass dehydration, acute kidney injury, and a transient reduction in renal function termed “initial dip”. Specifically, in situations where SGLT2is were introduced earlier in the observation period (from January 2014 to September 2021), the treatment was often discontinued prematurely at the discretion of the physicians due to reduced renal function, as there was a lack of widespread recognition of the ‘initial dip’ phenomenon at that time. A post hoc analysis of the EMPA-REG outcome revealed that 28.3% and 13.4% of the study participants (6668 patients) experienced > 10% decline in eGFR four weeks after the initiation of empagliflozin and a placebo, respectively. Patients with more advanced kidney disease and/or on diuretic therapy at baseline were more likely to undergo an initial eGFR dip of >10%. However, this initial dip did not have a major influence on the renal or cardiovascular benefits that empagliflozin was expected to offer [25].

Euglycemic ketoacidosis is a well-known adverse effect of SGLT2is [26]. In this study, three patients experienced diabetic ketoacidosis; notably, none were euglycemic. In certain instances, SGLT2i treatment was terminated due to weight loss. Importantly, weight loss is generally considered a favorable outcome, particularly for obese patients with diabetes [27]. In this study, the phrase “weight loss” as a reason for discontinuation encompasses a range of scenarios. Specifically, some patients were content with their diminished weight and sought to discontinue the SGLT2is owing to financial constraints. Conversely, others were already relatively lean, and upon initiating the SGLT2is, their weight declined to a degree that led physicians to view it as a risk factor for sarcopenia.

Furthermore, our findings suggested that older adults with diabetes tended to discontinue SGLT2i therapy. Regarding frequent urination in older individuals, overactive bladder and benign prostatic hyperplasia (the latter occurs only in males) are well-known to increase with age, both of which can trigger lower urinary tract symptoms, including frequent urination [28,29]. Moreover, frequent urination often involves nocturia, which not only deteriorates the quality of life but also increases the risk of falling, subsequently restricting activities of daily living, especially among older individuals [30]. Therefore, it is crucial to implement necessary precautions against frequent urination before initiating SGLT2is. Potential precautions include salt restriction, appropriate advice on water intake, and selecting an SGLT2i with a relatively short biological half-life. Reportedly, excessive salt intake may induce frequent urination and nocturia; hence, salt restriction is a potential precaution [31,32,33]. A randomized control trial is underway to examine the effect of salt restriction on nocturia under SGLT2i medication in patients with type 2 diabetes [34]. When prescribed SGLT2is for the first time, patients are often advised to drink more than usual amounts of water to avoid dehydration. However, this advice can lead to excessive water intake after the culmination of the initial diuretic effect of SGLT2is, because a previous report indicated that the increase in urine volume due to SGLT2is occurred early in the course of administration and was virtually unchanged one month after administration [35]. In fact, the frequency of discontinuation of SGLT2is due to frequent urination was not different between the durations of SGLT2i treatment in our study. Thus, advice on water intake should be tailored according to the patient’s age and the duration for which the SGLT2i has been administered. Selecting an SGLT2i with a relatively short biological half-life could minimize the night-time effects of osmotic diuresis, as SGLT2is are normally administered every morning. In the current study, we could not compare the degree of frequent urination among the SGLT2is owing to the small sample size. Notably, a study comparing tofogliflozin (t_1/2_: 5.4 h) with ipragliflozin (t_1/2_: 14.97 h) has revealed lower nocturnal urinary glucose, urinary frequency, and urinary volume in the tofogliflozin-treated group than those in the ipragliflozin-treated group [36].

In summary, particularly in older patients who are at an elevated risk of frequent urination, physicians must exercise increased vigilance in assessing the presence of lower urinary tract symptoms prior to initiating SGLT2i therapy. If such symptoms are present, treatment for the underlying diseases should precede. Moreover, implementing salt restriction, offering tailored advice on water intake, and opting for an SGLT2i with a shorter biological half-life would be advantageous.

Our study has some limitations. First, given that our study was retrospective, we could not assess the causality between each factor and SGLT2i discontinuation. Second, the decision to discontinue SGLT2i treatment depended on the physician. Third, concomitant use of diuretics, other antihypertensive drugs, and antidiabetic drugs with SGLT2is may alter the risk of polyuria and hypotension and, ultimately, the discontinuation of SGLT2is. However, data on administering such drugs at baseline were unavailable. Although data on insulin use were available, insulin use did not influence the frequency of discontinuing SGLT2i treatment. Fourth, comparisons among different types of SGLT2 inhibitors would be beneficial, but some of them were too small in sample size to perform the analysis.

In conclusion, this real-world study revealed that frequent urination was the most common reason for discontinuing treatment with SGLT2is. To avoid SGLT2i discontinuation, precautions must be taken against various factors that may result in frequent urination. Further studies are required to confirm the effectiveness of these precautions, particularly among older patients.

## Figures and Tables

**Table 1 jcm-12-06993-t001:** Patient characteristics at registration.

n	766
Age (years)	64 [54–71]
Sex (male) n, (%)	441 (57.6)
Duration (years), (n = 672)	11 [4–18]
Type 1 diabetes n, (%)	53 (6.9)
Body mass index (kg/m^2^)	25.3 [23.0–28.3]
HbA1c (%)	7.9 [7.1–8.7]
Blood glucose (mg/dL)	157.0 [127.0–199.0]
Systolic blood pressure (mmHg), (n = 691)	131.0 [122.0–143.0]
Diastolic blood pressure (mmHg), (n = 690)	76.0 [69.0–85.0]
Creatinine (mg/dL), (n = 757)	0.75 [0.61–0.91]
Estimated glomerular filtration rate (mL/min per 1.73 m^2^), (n = 757)	74.5 [61.0–86.7]
Total cholesterol (mg/dL), (n = 705)	195.0 [166.0–223.0]
High-density lipoprotein cholesterol (mg/dL), (n = 729)	53.0 [45.0–64.0]
Triglycerides (mg/dL), (n = 738)	140.0 [97.0–213.8]
Insulin (+/−) n, (%)	175/591 (22.8/77.2)
SGLT2i (dapagliflozin/canagliflozin/luseogliflozin/ empagliflozin/ipragliflozin/tofogliflozin) n, (%)	414/131/92/61/36/32 (54.0/17.1/12.0/8.0/4.7/4.2)
Smoking(never/past/current) n, (%), (n = 679)	322/240/117 (47.4/35.3/17.2)
Neuropathy (+/−) n, (%), (n = 563)	122/441 (21.7/78.3)
Retinopathy (NDR/SDR/PPDR or PDR) n, (%), (n = 766)	531/80/155 (69.3/10.4/20.2)
Nephropathy (stage 1/2/3/4 or 5) n, (%), (n = 766)	474/209/77/6 (61.9/27.3/10.0/0.8)
Cardiovascular disease (+/−) n, (%), (n = 766)	93/673 (12.1/87.9)

Continuous variables are expressed as median (interquartile range). NDR, normal diabetic retinopathy; SDR, simple diabetic retinopathy; PPDR, preproliferative diabetic retinopathy; PDR, proliferative diabetic retinopathy.

**Table 2 jcm-12-06993-t002:** The reasons for discontinuation of the SGLT2 inhibitors (SGLT2is) and their stratification by the duration of the SGLT2i treatment.

	Total (n = 97)	<3 Months (n = 22)	3 to 12 Months (n = 43)	12 to 24 Months (n = 32)
Frequent urination	19 (19.6%)	4 (18.2%)	7 (16.2%)	8 (22.8%)
Genital infection	11 (11.3%)	2 (9.1%)	3 (9.3%)	6 (17.1%)
Renal dysfunction	8 (8.2%)	2 (9.1%)	5 (11.6%)	1 (2.9%)
Urinary tract infection	7 (7.2%)	0 (0%)	5 (11.6%)	2 (5.7%)
Fatigue	4 (4.1%)	2 (9.1%)	2 (4.7%)	0 (0%)
Digestive symptoms	4 (4.1%)	0 (0%)	3 (7.0%)	1 (2.9%)
Body weight loss	5 (5.2%)	2 (9.1%)	2 (4.7%)	1 (2.9%)
Diabetic ketoacidosis	3 (3.1%)	0 (0%)	2 (4.7%)	1 (2.9%)
Drug allergy	3 (3.1%)	1 (4.5%)	2 (4.7%)	0 (0%)
Unknown	23 (23.7%)	8 (36.4%)	7 (16.3%)	8 (22.9%)
Improved glycemic control	10 (10.6%)	1 (4.5%)	5 (11.6%)	4 (17.1%)

**Table 3 jcm-12-06993-t003:** The reasons for discontinuation of the SGLT2 inhibitors (SGLT2is) and their stratification by duration of the SGLT2i treatment in patients with type 2 diabetes.

	Total (n = 90)	<3 Months (n = 20)	3 to 12 Months (n = 39)	12 to 24 Months (n = 31)
Frequent urination	17 (18.9%)	4 (20.0%)	5 (12.8%)	8 (25.8%)
Genital infection	11 (12.2%)	2 (10.0%)	3 (7.7%)	6 (19.4%)
Renal dysfunction	7 (7.8%)	1 (5.0%)	5 (12.8%)	1 (3.2%)
Urinary tract infection	7 (7.8%)	0 (0%)	5 (12.8%)	2 (6.5%)
Fatigue	4 (4.4%)	2 (10.0%)	2 (5.1%)	0 (0%)
Digestive symptoms	4 (4.4%)	0 (0%)	3 (7.7%)	1 (3.2%)
Body weight loss	4 (4.4%)	1 (5.0%)	2 (5.1%)	1 (3.2%)
Diabetic ketoacidosis	1 (1.1%)	0 (0%)	1 (2.6%)	0 (0%)
Drug allergy	3 (3.3%)	1 (5.0%)	2 (5.1%)	0 (0%)
Unknown	22 (24.4%)	8 (40.0%)	6 (15.4%)	8 (25.8%)
Improved glycemic control	10 (11.1%)	1 (5.0%)	5 (12.8%)	4 (12.9%)

**Table 4 jcm-12-06993-t004:** Comparison of the continuation group with the discontinuation group.

	Continuation (n = 669)	Discontinuation (n = 87)	*p*
Age (years)	64 [53–71]	68 [55–75]	0.003
Sex (male/female) n, (%)	391/278 (58.4/41.6)	44/43 (50.6/49.4)	0.168
Duration (years)	11 [4–18] (n = 592)	9 [4–18] (n = 73)	0.54
Type 1/Type 2 diabetes n, (%)	46/623 (6.9/93.1)	7/80 (8.0/92.0)	0.832
Body mass index (kg/m^2^)	25.5 [23.2–28.3]	24.8 [21.2–28.8]	0.232
HbA1c (%)	7.9 [7.1–8.7]	7.8 [7.3–8.9]	0.467
Blood glucose (mg/dL)	155 [127–198]	172 [126–207.5]	0.096
Systolic blood pressure (mmHg)	131 [122–143] (n = 605)	131 [124–142] (n = 78)	0.389
Diastolic blood pressure (mmHg)	76 [69–84] (n = 604)	76 [69–85] (n = 78)	0.717
Creatinine (mg/dL)	0.76 [0.61–0.91] (n = 662)	0.75 [0.60–0.93] (n = 85)	0.95
Estimated glomerular filtration rate (mL/min per 1.73 m^2^)	74.8 [61.0–86.8] (n = 662)	69.9 [57.7–83.9] (n = 85)	0.194
Total cholesterol (mg/dL)	195 [166.0–223.9] (n = 618)	193 [165.0–215.5] (n = 79)	0.598
High-density lipoprotein cholesterol (mg/dL)	53 [45–63] (n = 637)	55 [46.0–65.9] (n = 83)	0.063
Triglycerides (mg/dL)	142 [99.0–214.0] (n = 645)	118 [83.8–198.3] (n = 84)	0.093
Insulin (+/−) n, (%)	516/153 (77.1/22.9)	66/21 (75.9/24.1)	0.787
Smoking (never/past/current) n, (%)	274/219/104 (45.9/36.7/17.4) (n = 597)	43/17/12 (59.7/23.6/16.7) (n = 72)	0.055
Neuropathy (+/−) n, (%)	115/388 (22.9/77.1) (n = 503)	7/47 (13.0/87.0) (n = 54)	0.118
Retinopathy (NDR/SDR/PPDR or PDR) n, (%)	470/70/129 (70.3/10.5/19.3) (n = 669)	55/9/23 (63.2/10.3/26.4) (n = 87)	0.268
Nephropathy (stage 1/2/3/4 or 5) n, (%)	416/182/66/5 (62.2/27.2/9.9/0.4) (n = 669)	52/23/11/1 (59.8/26.4/12.6/1.1) (n = 87)	0.664
Cardiovascular disease (+/−) n, (%)	80/589 (12.0/88.0) (n = 669)	13/74 (14.9/85.1) (n = 87)	0.391

Continuous variables are expressed as median (interquartile range). Patients who discontinued SGLT2is because of improved glycemic control are excluded from this table. NDR, normal diabetic retinopathy; SDR, simple diabetic retinopathy; PPDR, preproliferative diabetic retinopathy; PDR, proliferative diabetic retinopathy.

## Data Availability

The datasets from our study are available on reasonable request through the corresponding author.

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
