# Peer review of "Reasons for Discontinuing Treatment with Sodium-Glucose Cotransporter 2 Inhibitors in Patients with Diabetes in Real-World Settings: The KAMOGAWA-A Study"

_jcm, 2023, doi:10.3390/jcm12226993_

Round 1

Reviewer 1 Report

Comments and Suggestions for Authors

My compliments with the authors for the paper presented, but introduction must be improved. It can be useful to add the "recent pharmacological options in type 2 diabetes and synergic mechanism in cardiovascular disease" in the introduction, it is really important underline the importance of this group of treatments for diabetes. The patients were included until 2021, in the patients that have discontiuned the study were in treatment only with metformin and SGLT2i? in patients that continued the SGLT2i was combined with insulin or other antidiabetic drug so the effect of urination was less present?

Comments on the Quality of English Language

minor spelling and grammatic check is necessary

Author Response

Dear Reviewer 1,

Thank you very much for your valuable comments. Point-by-point responses to your comments and suggestions are made below.

My compliments with the authors for the paper presented, but introduction must be improved. It can be useful to add the "recent pharmacological options in type 2 diabetes and synergic mechanism in cardiovascular disease" in the introduction, it is really important underline the importance of this group of treatments for diabetes. The patients were included until 2021, in the patients that have discontiuned the study were in treatment only with metformin and SGLT2i? in patients that continued the SGLT2i was combined with insulin or other antidiabetic drug so the effect of urination was less present?

Response

As you say, it is crucial to emphasize the significance of synergic mechanism in cardiovascular disease. We have already mentioned that fact in Lines 43-44, and 46-47.

Various classes of antidiabetic drugs are used in our institutions, but unfortunately, we have no data available on the administration of other antidiabetic drugs at baseline. The data on the use of insulin are available, so we added the number of insulin users to Table 1 and Table 3. In this study, the use of insulin did not influence the frequency of discontinuing SGLT2i treatment.

According to your suggestion, we have described as below in the Limitation section.

“Third, concomitant use of diuretics, other antihypertensive drugs and antidiabetic drugs with SGLT2is may alter the risk of polyuria and hypotension and ultimately the discontinuation of SGLT2is. However, the data on the administration of such drugs at baseline are not available. Although the data on the use of insulin are available, the use of insulin did not influence the frequency of discontinuing SGLT2i treatment.”

Reviewer 2 Report

Comments and Suggestions for Authors

The authors present data from a real-world observational study on reasons for discontinuation of SGLT2i treatments in patients with T2D.

The overall rationale of the study clear, even though side effects of SGLT2i are well known in character in frequency.

Introduction: Fine.

Methods: Excluding patients with follow-up < 2 years excludes all patients with early fatal events, including UTI-related sepsis, ketoacidosis, syncopal accidents by hypotension and so on. All discontinuations and fatal events of that group should be included in the analysis.

Results: Generally fine.

Please report the frequency of individual SGLT2i substances, as the prescription policy may be specific for Japan.

Please report the rate of co-medication with diuretics and other antihypertensive drugs, as they may increase the risk for polyuria and hypotension.

Discussion:
Both given limitations are misleading. The study is prospective or retrospective, but not cross-sectional, as it covers 2 years of observation time. Reasons of discontinuation do not depend on physicians. The decision to discontinue may have.

Comments on the Quality of English Language

minor changes needed

Author Response

Dear Reviewer 2,

Thank you very much for your valuable comments. Point-by-point responses to your comments and suggestions are made below.

Methods: Excluding patients with follow-up < 2 years excludes all patients with early fatal events, including UTI-related sepsis, ketoacidosis, syncopal accidents by hypotension and so on. All discontinuations and fatal events of that group should be included in the analysis.

Response

The patients who were excluded due to shortage of follow-up time were in the continuation group, not in the discontinuation group. Therefore, we have already included all patients with early fatal events you mention above. We have failed to make it clear on that. We modified the sentence in the Methods section as follows: “The patients in the continuation group whose follow-up period were less than 2 years were also excluded (n=52).”

Results: Generally fine.

Please report the frequency of individual SGLT2i substances, as the prescription policy may be specific for Japan.

Response

In our study, the frequency of individual SGLT2 inhibitors are as follows: dapagliflozin (414 patients, 54.0%), canagliflozin (131 patients, 17.1%), luseogliflozin (92 patients, 12.0%), ipragliflozin (36 patients, 4.7%), tofogliflozin (32 patients, 4.2%).

According to your suggestion, we will report the frequency of individual SGLT2 inhibitors by adding the information above to Table 1.

Please report the rate of co-medication with diuretics and other antihypertensive drugs, as they may increase the risk for polyuria and hypotension.

Response

As you say, concomitant use of diuretics and other antihypertensive drugs may alter the risk of polyuria and hypotension. However, we have no data on the administration of diuretics nor other antihypertensive drugs at baseline.

We added to the Limitation section, “Third, concomitant use of diuretics, other antihypertensive drugs and antidiabetic drugs with SGLT2is may alter the risk of polyuria and hypotension and ultimately the discontinuation of SGLT2is. However, the data on the administration of such drugs at baseline are not available. Although the data on the use of insulin are available, the use of insulin did not influence the frequency of discontinuing SGLT2i treatment.”

Discussion:

Both given limitations are misleading. The study is prospective or retrospective, but not cross-sectional, as it covers 2 years of observation time. Reasons of discontinuation do not depend on physicians. The decision to discontinue may have.

Response

As you have pointed out, our study is retrospective, and the decision to discontinue SGLT2is may depend on physicians, not the reasons for discontinuation. We corrected the sentences in the Limitation section as follows; “First, given that our study was retrospective, we could not assess the causality between each factor and SGLT2i discontinuation. Second, the decision to discontinue SGLT2i treatment depended on the physician.”

Reviewer 3 Report

Comments and Suggestions for Authors

The paper observed the causes of SGLT2i discontinuation in patients with type 2 DM. Abstract, background, methods and presentation of the results are fine. However, discussion is rather short. I suggest authors to compare their results with literature data. For example, urinary tract infection is associated with SGLT2i treatment (Clin Case Rep. 2023;11(1):e6803. doi: 10.1002/ccr3.6803). In addition, euglycaemic ketoacidosis is a well established complication of SGLT2i drugs (J Coll Physicians Surg Pak. 2022 Jul;32(7):928-930. doi: 10.29271/jcpsp.2022.07.928). Authors also found that weight loss was a cause of drug discontinuation. However, weight loss is a desirable effect of these drugs and probably associated with well diabetic control (Ir J Med Sci. 2022;191(4):1647-1652. doi: 10.1007/s11845-021-02761-6). Finally, reduced renal function was another cause of SGLT2i discontinuation in present work but literature data suggest that these drugs are associated with improvement in diabetic kidney disease ((Front Endocrinol (Lausanne). 2023 Jan 26;14:1026040. doi: 10.3389/fendo.2023.1026040), and (Int J Mol Sci. 2022 Nov 9;23(22):13749. doi: 10.3390/ijms232213749)). 

Another suggestion for discussion section could be that authors may consider commenting on clinical translation of the study results. What will those results bring about in daily clinical practice?

Author Response

Dear Reviewer 3,

Thank you very much for your valuable comments. Point-by-point responses to your comments and suggestions are made below.

I suggest authors to compare their results with literature data. For example, urinary tract infection is associated with SGLT2i treatment (Clin Case Rep. 2023;11(1):e6803. doi: 10.1002/ccr3.6803). In addition, euglycaemic ketoacidosis is a well established complication of SGLT2i drugs (J Coll Physicians Surg Pak. 2022 Jul;32(7):928-930. doi: 10.29271/jcpsp.2022.07.928).

Response

As you say, more comparisons with literature data are important.

A meta-analysis showed that urinary tract infection was associated with dapagliflozin, but not with other SGLT2is (Diabetes Obes Metab. 2017 Mar;19(3):348-355.). In this study, a total of 7 patients discontinued SGLT2i treatment because of urinary tract infection, of whom 5 are administered dapagliflozin, and 2 luseogliflozin. However, as this study is retrospective, this result does not prove causal relationship.

As you mentioned, euglycemic ketoacidosis is a well-known adverse effect of SGLT2is. In this study, 3 patients are affected with diabetic ketoacidosis, but none of them were euglycemic.

According to your suggestion, we added to the Discussion section, “A meta-analysis showed that urinary tract infection was associated with dapagliflozin, but not with other SGLT2is [18]. In this study, a total of 7 patients discontinued SGLT2i treatment because of urinary tract infection, of whom 5 were administered dapagliflozin, and 2 luseogliflozin. However, as this study is retrospective, this result does not prove causal relationship.”

“Euglycemic ketoacidosis is a well-known adverse effect of SGLT2is [21]. In this study, 3 patients were affected with diabetic ketoacidosis, but none of them were euglycemic.”

Authors also found that weight loss was a cause of drug discontinuation. However, weight loss is a desirable effect of these drugs and probably associated with well diabetic control (Ir J Med Sci. 2022;191(4):1647-1652. doi: 10.1007/s11845-021-02761-6).

Response

As you say, weight loss is a desirable effect especially for obese patients. In our study, the term “weight loss” as a cause of discontinuation encompasses various situations. Some patients were satisfied with their reduced weight and hoped to discontinue SGLT2is due to financial burden, while others were relatively lean, and by initiating SGLT2is their weight reduced to the extent that physicians started to perceive it as a risk for sarcopenia.

According to your suggestion, we added to the Discussion section, “In some cases, SGLT2i treatment was discontinued because of weight loss. Weight loss is essentially a desirable effect [22] especially for obese patients with diabetes. In this study, the term “weight loss” as a cause of discontinuation encompasses various situations. Some patients were satisfied with their reduced weight and hoped to discontinue SGLT2is due to financial burden, while others were relatively lean, and by initiating SGLT2is their weight reduced to the extent that physicians started to perceive it as a risk for sarcopenia.”

Finally, reduced renal function was another cause of SGLT2i discontinuation in present work but literature data suggest that these drugs are associated with improvement in diabetic kidney disease ((Front Endocrinol (Lausanne). 2023 Jan 26;14:1026040. doi: 10.3389/fendo.2023.1026040), and (Int J Mol Sci. 2022 Nov 9;23(22):13749. doi: 10.3390/ijms232213749)).

Response

As you say, SGLT2is are associated with improvement in diabetic kidney disease which makes continuation of SGLT2i treatment more important. First of all, the cases of reduced renal function include dehydration, acute kidney injury and transient reduction of renal function called “initial dip”. As for initial dip, especially in cases that SGLT2is were initiated earlier in the observation period (From January 2014 to September 2021), SGLT2i treatment was discontinued early at physicians’ discretion because of reduced renal function as the concept of initial dip had not been widespread.

According to your suggestion, we added to the Discussion section, “SGLT2is are associated with improvement in diabetic kidney disease [19], which makes continuation of SGLT2i treatment more significant. The cases of reduced renal function in this study include dehydration, acute kidney injury and transient reduction of renal function called “initial dip”[20]. As for initial dip, especially in cases that SGLT2is were initiated earlier in the observation period (From January 2014 to September 2021), SGLT2i treatment was discontinued early at physicians’ discretion because of reduced renal function as the concept of initial dip had not been widespread.”

Another suggestion for discussion section could be that authors may consider commenting on clinical translation of the study results. What will those results bring about in daily clinical practice?

Response

As you say, commenting on clinical translation of this study is important. According to your suggestion, we added to the Discussion section, “Especially in older patients, who are at higher risk of frequent urination, physicians have to be more attentive to whether the patients already have lower urinary tract symptoms before initiating SGLT2i treatment. If the symptoms are present, the treatment of diseases causing them should precede, and salt restriction, appropriate advice on water intake selecting an SGLT2i that has shorter biological half-life would be beneficial.”

Reviewer 4 Report

Comments and Suggestions for Authors

This real-world data analysis from the KAMOGAWA-A Cohort provides new insights into the practical challenges of managing patients on SGLT2 inhibitors, suggesting that frequent urination was the most common reason for discontinuing treatment with SGLT2i. To avoid SGLT2i discontinuation, precautions must be taken against various factors that may result in frequent urination. Overall, the manuscript is very well written. The experimental approach is straightforward and clearly described. Therefore, I only have a few minor comments to improve this manuscript.

1.  A more in-depth comparison of different type of SGLT2 inhibitors, potentially via a larger study or meta-analysis, would be beneficial.

2. More explicit guidance for clinicians on patient counseling, lifestyle recommendations, and SGLT2i selection would enhance the manuscript's utility.

Author Response

Dear Reviewer 4,

Thank you very much for your valuable comments. Point-by-point responses to your comments and suggestions are made below.

  1. A more in-depth comparison of different type of SGLT2 inhibitors, potentially via a larger study or meta-analysis, would be beneficial.

Response

As you say, comparisons between each type of SGLT2 inhibitors would be beneficial, but some of them are too small in sample size to perform analysis. If we refer to larger studies, post-marketing surveillance studies of SGLT2 inhibitors have consistently reported that frequent urination, genital infection, and urinary tract infection were major adverse drug reactions, although it did not necessarily mean that those reactions led to discontinuation of SGLT2i treatment. This study is novel in a sense that it provides new insight into what adverse reactions are prone to discontinuation of SGLT2i treatment.

According to your suggestion, we added to the Discussion section, “If we refer to other larger studies, post-marketing surveillance studies of SGLT2 inhibitors have consistently reported that frequent urination, genital infection, and urinary tract infection were major adverse drug reactions, although it did not necessarily mean that those reactions led to discontinuation of SGLT2i treatment [14-17]. This study is novel in a sense that it provides new insight into what adverse reactions are prone to discontinuation of SGLT2i treatment.”, and to the Limitation section, “Fourth, comparisons among different types of SGLT2 inhibitors would be beneficial, but some of them are too small in sample size to perform analysis.”

  1. More explicit guidance for clinicians on patient counseling, lifestyle recommendations, and SGLT2i selection would enhance the manuscript's utility.

Response

As you say, more explicit clinical guidance would make our manuscript easier to utilize. We have added to the Discussion section, “In summary, especially in older patients, who are at higher risk of frequent urination, physicians have to be more attentive to whether the patients already have lower urinary tract symptoms before initiating SGLT2i treatment. If the symptoms are present, the treatment of diseases causing them should precede, and salt restriction, appropriate advice on water intake, and selecting an SGLT2i that has shorter biological half-life would be beneficial.”

Round 2

Reviewer 2 Report

Comments and Suggestions for Authors

Revision was successful, I consider the paper ready for acceptance.

Comments on the Quality of English Language

minor changes needed
